# Characterization of Two *Zymomonas mobilis* Wild Strains and Analysis of Populations Dynamics during Their Leavening of Bread-like Doughs

**DOI:** 10.3390/foods11182768

**Published:** 2022-09-08

**Authors:** Claudia Picozzi, Elisa Clagnan, Alida Musatti, Manuela Rollini, Lorenzo Brusetti

**Affiliations:** 1Department of Food, Environmental and Nutritional Sciences (DeFENS), Università degli Studi di Milano, Via Celoria 2, 20133 Milano, Italy; 2Faculty of Science and Technology, Free University of Bozen/Bolzano, Piazza Università 5, 39100 Bolzano, Italy

**Keywords:** *Zymomonas mobilis*, dough leavening, ARISA analysis, population dynamics, lag leavening time

## Abstract

Two *Zymomonas mobilis* wild strains (UMB478 and 479) isolated from water kefir were characterized for their biomass production levels and leavening performance when used as the inoculum of a real bread-like dough formulation. The obtained baked product would be consumable by people with adverse responses to *Saccharomyces cerevisiae*. In liquid cultures, the two strains reached similar biomass concentration (0.7 g CDW/L). UMB479 showed an interesting resistance to NaCl (MBC 30 g/L), that may be useful in the bakery sector. When inoculated in doughs, UMB479 produced the maximum dough volume (650 mL) after 5 h, glucose was almost consumed and 1 g/100 g of ethanol produced, +200% respective to UMB478. Using *S. cerevisiae* for comparison purposes, the dough doubled its volume fast, in only 2 h, but reached a final level of 575 mL, lower than that achieved by *Z. mobilis*. The analysis of bacterial and fungal population dynamics during dough leavening was performed through the Automated Ribosomal Intergenic Spacer Analysis (ARISA); doughs leavened by UMB479 showed an interesting decrease in fungal richness after leavening. *S. cerevisiae,* instead, created a more complex fungal community, similar before and after leavening. Results will pave the way for the use of *Z. mobilis* UMB479 in commercial yeast-free leavened products.

## 1. Introduction

Bread represents a fundamental food in many societies; in its simplest form, bread is made using flour, baker’s yeast (*Saccharomyces cerevisiae*), salt and water as ingredients. Human exposure to baker’s yeast is frequent due to the massive consumption of bread but also of alcoholic beverages (beer, cider, wine) and even nutraceuticals and food supplement use [1]. So far, there are concerns that the incidence of yeast intolerance is increasing in Western countries.

The possibility of adverse responses related to baker’s yeast ingestion is mainly spread across patients affected by Inflammatory Bowel Disease (IBD), Crohn’s disease (CD) and other autoimmune disorders; in these cases, anti-*S. cerevisiae* antibodies (ASCA) directed against yeast phosphopeptidomannan can be used as specific diagnostic markers [2,3]. Recently, some researchers [4] stated that cases of yeast allergy in medical literature may be underestimated and should be considered in patients with adverse reactions to yeast-containing products. Among this population, a baker’s yeast-exclusion diet has to be proposed for the management of these patients, with immediate stabilization of their manifestations [5].

In this context, the opportunity of replacing *S. cerevisiae* with microorganisms possessing similar metabolic pathways is of utmost importance. The use of *Zymonomas mobilis* as a leavening agent may increase the variety of baked goods alternatives to those leavened by yeast. The potential product will be addressed to people having adverse responses to the ingestion of baked foods leavened with *S. cerevisiae*, providing a solution to the growing incidence of yeast hypersensitivity. This bacterium is able to metabolize only glucose, fructose and sucrose but not maltose through the Entner-Doudoroff (ED) pathway; in comparison with glycolysis, the ED route yields 1 ATP instead of 2 per every glucose molecule processed, as well as 1 NADH and 1 NADPH instead of 2 NADH. Nevertheless, the enzymes involved in alcoholic fermentation are analogous to those present in yeasts, therefore resulting in the production of an equimolar mixture of ethanol and CO_2_ [6]. Thanks to its high ethanol resistance (up to 12%), *Z. mobilis* is also gaining great interest in bioethanol production [6,7].

Musatti et al. [8] reported that *Z. mobilis* strains belonging to the official DSMZ collection (Deutsche Sammlung von Mikroorganismen und Zellkulturen GmbH), grown on a medium without yeast–derived compounds, were able to efficiently leaven model doughs when glucose was added to boost the ED catabolic pathway (20 g flour, 15 mL distilled water and 2 g glucose, leavening at 29 °C). Its interesting leavening performance was also confirmed in bread-like dough formulations (333 g flour, 167 mL distilled water) added with glucose or fructose [9,10]. However, when employing these strains, the addition of NaCl was found to slow down the dough rising. Doughs leavened by *Z. mobilis* DSMZ strains were also analyzed for their metabolomic profiles of the aromatic VOCs developed during leavening. This bacterial species developed proper fermentation products, such as ethanol, acetic acid and 2-butanone-4-hydroxy, similarly to traditional starter cultures used in bakery applications. However, these strains were also able to address a unique signature in fermented doughs for the production of nonanoic and undecanoic acids [11].

In the present research, we focused on the use of two *Z. mobilis* wild strains, isolated by water kefir. Water kefir is a slightly carbonated, acidic beverage produced by fermenting a solution of sucrose, to which figs or other dried fruits and lemon have been added. The fermentation process is generally driven by a combination of bacteria and yeasts which live in symbiosis in water kefir grains.

These two wild strains were first characterized for their biomass production levels in the absence or increasing presence of NaCl; successively, their leavening performance was tested in a real bread-like dough formulation, containing 1.5% NaCl (*w*/*w* on flour). The analysis of the population dynamics of bacterial and fungal communities during dough leavening was also performed through the Automated Ribosomal Intergenic Spacer Analysis (ARISA). This characterization allowed for changes to be monitored inside the microbial communities present in doughs over leavening time. Results were compared with data obtained from doughs leavened with *S. cerevisiae*, in order to understand how the presence of *Z. mobilis* may affect the dough’s microbial population and richness. The obtained results will pave the way for the safe use of selected wild strains of *Z. mobilis* in commercial yeast-free salty-leavened products.

## 2. Materials and Methods

### 2.1. Microorganisms and Maintenance

In the present study, two new isolates of *Z. mobilis* were used i.e., UMB 478 and UMB 479 (UMB: University of Milan Bacterial Collection). Strains were maintained in DSM liquid medium [8]. All ingredients were dissolved in distilled water, pH set at 6.8 and sterilized at 112 °C for 30 min. Stock cultures of both microorganisms were stored at −80 °C in the same medium with 20% (*v*/*v*) glycerol (VWR International, Leuven, Belgium).

Samples of compressed commercial baker’s yeast (*S. cerevisiae*) were purchased at a local supermarket (Carrefour, Milan, Italy). Yeast was stored at 4 °C and used within the first two weeks after purchase.

### 2.2. Z. mobilis Growth Kinetics

Cell growth of *Z. mobilis* was studied at 30 °C in 384-well plates containing DSM liquid medium with different percentages of inoculum, i.e., 5 and 10% (*v*/*v*); for both strains, glucose, fructose and sucrose (20 g/L) were alternatively used as carbon and energy sources, while maintaining the other ingredients of DSM medium (bacto-peptone and yeast extract) without changing the pH. Each condition was tested in five replicates.

Culture turbidity (OD) was measured spectrophotometrically at 600 nm every 30 min for 72 h in a PowerWave™ XS2 Microplate Spectrophotometer (BioTek, Winooski, VT, USA); lag phase (min) and maximum growth rate (OD/h) were determined by fitting data using the DMFit 3.5 Excel add-in [12].

### 2.3. Z. mobilis Biomass Production

The two strains were cultured in 1 L flasks with 600 mL of DSM-YF (yeast-free) liquid medium containing glucose, casein hydrolysate (10 g/L) and meat extract (5 g/L). Flasks were inoculated with 5% (*v*/*v*) of an 8 h pre-culture grown in DSM medium. Cultures were incubated at 30 °C for 16 h under stationary conditions.

Biomass growth was determined in terms of cell dry weight (CDW), as previously reported [8]. Briefly, an aliquot of culture broth was centrifuged at 8600× *g* for 20 min at 5 °C in a JA-10 centrifuge (Beckman Coulter, Cassina de Pecchi, Milan, Italy), and the recovered cell pellet was washed with deionized water and centrifuged again. Cell dry weight (CDW) was determined after drying the cell pellet at 105 °C for 18–24 h.

### 2.4. Z. mobilis Sensitivity to NaCl: MIC and MBC

*Z. mobilis* cell growth was evaluated at 30 °C in 96-well plates containing DSM liquid cultures (10% *v*/*v* inoculum) and different NaCl concentrations (Merck KGaA, Darmstadt, Germany): 0–1.25–2.5–5–10–15–20–30–40 g/L. Culture turbidity (OD) was measured spectrophotometrically at 600 nm every 15 min for 24 h, as reported in Section 2.2. The lag phase (min) and maximum growth rate (OD/h) were determined by fitting data on DMFit 3.5 Excel add-in [12]. Minimum Inhibitory Concentration (MIC) was determined as the lowest NaCl concentration that produced no visible growth, i.e., an OD not significantly different from that of non-inoculated wells. For the evaluation of Minimum Bactericidal Concentration (MBC), 10 µL of culture collected at the end of the analysis were inoculated on DSM agar plates (DSM added with 15 g/L of agar) and incubated for 24–48 h at 30 °C. MBC was determined as the lowest NaCl concentration producing less than 10 colonies on plates.

### 2.5. Doughs Production

Doughs were prepared with 288 g of 00 wheat flour (Molino Colombo, Paderno d’Adda, Lecco, Italy), glucose 8.64 g (3% *w*/*w* on flour), NaCl 4.32 g (1.5% *w*/*w* on flour) and adding 158 mL of distilled water in which the inoculum (*Z. mobilis* or *S. cerevisiae*) was suspended in terms of 1.5 mg CDW/g of dough (5 × 10^8^ cells/g dough). The amount of water was determined through the Brabender^®^ Farinograph (Brabender OHG, Duisburg, Germany; 300 g chamber, 30 °C) in order to obtain a dough consistency of 500 ± 25 BU (Brabender Unit). Ingredients were mixed in a blender (CNUM5ST, Bosch, Stuttgart, Germany) at speed 1 for 1 min and at speed 2 for 3 min.

The dough was divided into 3 sections [10] as follows:
-250 g, inserted into a 1 L graduate cylinder to evaluate the dough volume increase every 30 min for up to 6 h of leavening.-25 g, inserted into a double chamber flask connected to a graduate burette filled with 0.1 M NaCl, to evaluate the total amount of CO_2_ produced every 30 min for up to 6 h of leavening. Two other unconventional indices were extrapolated from the dough development curves: the lag leavening time (LLT, h), defined as the time before an increase in dough height was noticed, the leavening rate (LR, mm/h), as the slope of the first linear part of the curve after LLT, and maximum dough volume (DV max, mL) [12].-The remaining sample was left to leaven in 4 closed sterile containers, one for each sampling time. The doughs were incubated at 30 °C and samples were taken at 0, 2, 4 and 6 h to determine dough pH and to perform microbiological and HPLC (High-Performance Liquid Chromatography) analyses.

### 2.6. Microbiological Analyses

At appropriate leavening intervals, 5 g of dough was diluted in 45 mL sterile peptone water (10 g/L Bacto-peptone in distilled H_2_O, pH 6.8) and homogenized in a Stomacher (Seward, Worthing West Sussex, UK) for 3–4 min. After appropriate decimal dilutions, suspensions were plated in proper media: *Z. mobilis* onto DSM agar, incubated at 30 °C for 3 days under anaerobic conditions; and baker’s yeast (*S. cerevisiae*) onto MEA (Malt Extract Agar, Oxoid, Basinstoke, UK), then incubated at 30 °C for 48 h. Total mesophilic aerobic bacterial count (TBC) was determined by pour plating on Tryptic Soy Agar (TSA, Scharlab, Barcelona, Spain) after incubation at 30 °C for 24–48 h. When *S. cerevisiae* was used, TSA was added with 0.1% (*v*/*v*) cycloheximide (Sigma-Aldrich) to avoid TBC overestimation due to yeast growth. Yeasts and molds were determined by plating on Yeast Glucose Chloramphenicol Agar (YGC-Scharlab, Barcelona, Spain) and incubated at 25 °C for 3–5 days. Counts were reported as logarithms of the number of colony-forming units (Log CFU/g of dough) with means and standard deviation values of two technological replicates.

### 2.7. Analytical Determinations

Consumption of sugars (maltose and glucose) as well as ethanol, lactic and acetic acid produced during leavening, were determined through an HPLC system (L 7000, Merck Hitachi, Darmstadt, Germany) equipped with RI and UV (210 nm) detectors connected in series as reported by Musatti et al. [13]. Aliquots of 2–4 mL of homogenized and appropriately diluted dough samples were centrifuged (Eppendorf 5804, 10,600× *g*, 10 min) and the obtained supernatants were filtered through a 0.45 µm syringe filter (VWR International, Leuven, Belgium) before HPLC analysis. Data were referred to as 100 g of flour (g/100 g). Results represent the mean and standard deviation values of two technological replicates.

Dough pH was monitored at different intervals for the whole undiluted dough sample (pH-meter mod. pH 510, Eutech, Toronto, ON, Canada).

### 2.8. Automated Ribosomal Intergenic Spacer Analysis (ARISA)

Dough aliquots were collected immediately after dough production (t0) and after a 6 h leavening (t6) for the characterization of the bacterial and fungal communities at the beginning and at the end of leavening. Total DNA was extracted in triplicate from each aliquot. Each replica was powdered in liquid nitrogen using a sterilized mortar and then DNA was extracted through the DNeasy PowerSoil kit (QIAGEN, Hilden, Germany) according to manufacturer guidance with the first vortexing step performed in an Eppendorf ThermoMixer Comfort (Eppendorf, Wesseling, Germany) at 1400 rpm for 10 min. The purified DNA yield was quantified using Qubit™ (Thermo Fisher Scientific, Waltham, MA, USA), while the quality was determined through gel electrophoresis of 10 g/L 1 × TAE agarose gels. DNA was stored at −80 °C until analyses.

Polymerase chain reactions (PCRs) for the fingerprinting of fungal and bacterial analyses were executed as per Table A1. PCR products were checked through gel electrophoresis 15 g/L 1 × TAE agarose gels. For ARISA analyses, 3 μL of each PCR reaction, including negative controls and controls containing the DNA from *Z. mobilis* and *S. cerevisiae* cultures, were mixed with 14 μL of HiDi Formamide (Applied Biosystems—Life Technologies, Carlsbad, CA, USA), plus 0.8 μL of 1200 bp LIZ standards (Molecular probes—Life Technologies, USA). Samples were shipped to STAB Vida Lda. (Caparica, Portugal) for capillary electrophoresis. Raw peak profiles were analyzed using the AB Peak Scanner Software™ v1.0 (Applied Biosystems, Monza, Italy). Peaks below a fluorescence threshold of 50 units were discarded from the analyses [14], while all fragments between 150 and 1400 bp were considered. The output table was exported and converted using Microsoft-Excel macro Treeflap [15]. The data matrix obtained through Treeflap was further normalized and only peaks present in all replicates were kept for further analyses to remove rare species and facilitate interpretation.

### 2.9. Statistical Analysis

Analytical results were treated with one-way analyses of variance (ANOVA) and when the effect was significant (*p* < 0.05), the differences between the means were separated by the Tukey-b test with multiple comparisons. Data were processed by Sigma Plot (v. 14, Systat Sofware, Inc., San Jose, CA, USA).

All statistical analyses were performed on R studio (version 4.1.1, Ihaka R. and Gentleman R., Auckland, Australia; www.r-project.com; accessed on 1 September 2021). Taxonomic summaries were performed using the phyloseq package [16]. Richness and diversity indexes were calculated using the ‘vegan’ package [17]. After the Shapiro-Wilk test was used to test for normality, differences were tested by one-way analysis of variance (ANOVA) followed by Tukey’s post hoc test (*p* < 0.05), while non-normal data were analyzed through a non-parametric Kruskal-Wallis test followed by Dunn’s Test. T-test and Wilcoxon-signed rank test were used for normal and non-normal pairwise comparisons, respectively.

Multivariate analyses were performed on the peak’s relative abundances. Non-metric multidimensional scaling (NMDS) was constructed based on Bray-Curtis distances. Results were confirmed through the PERMANOVA test [17]. Furthermore, pairwise comparisons were performed with the ‘pairwiseAdonis’ package [18].

## 3. Results and Discussion

### 3.1. Z. mobilis Growth and Biomass Production

Cell growth of *Z. mobilis* was studied in cultures having alternatively used glucose, fructose or sucrose as carbon and energy source. The parametrization was carried out on time courses of culture turbidity by applying the DmFit Excel add-in to obtain the following indices: lag phase duration (min), maximum growth rate (OD/h) and final OD reached (Y_end_) (Table 1). The two strains presented a similar behavior: the shortest lag phase durations (2.60 ± 0.42–2.76 ± 0.18 h) were evidenced using the highest cell inoculum (10% *v*/*v*) and glucose as carbon and energy source both for UMB478 and UMB479; the longest and statistically different (*p* < 0.05) lag phase was instead obtained with 5% cell inoculum for UMB478 with fructose (around 6.06 ± 0.20 h). The maximum growth rate was found to be not statistically different (*p* < 0.05) between strains, with performances in the range of 0.08–0.14 OD/h; nevertheless, both strains have statistically lower growth rates with fructose as substrate. The highest Y_end_ values (1.023 ± 0.041 OD for UMB478 and 1.078 ± 0.012 OD for UMB479, respectively) were again evidenced at 10% cell inoculum with sucrose for both strains. The higher growth parameters found for *Z. mobilis* with glucose may be explained by the fact that this bacterium is unique among prokariotes in that it relies solely on a facilitated diffusion system (uniport, GLF) to transport glucose and fructose without expending metabolic energy; however, GLF has a higher affinity (K_m_ 2–4 mM) for glucose than for fructose (K_m_ 40 mM). Sucrose, instead, is converted into glucose and fructose using up to three sucrose-splitting enzymes: an extracellular levansucrase (LevU), forming the fructose oligosaccharide levan and glucose; an extracellular invertase (InvB), forming glucose and fructose; and a second invertase (InvA), whose exact role and location in *Z. mobilis* is still unclear [19].

The two strains were then cultured in 1 L flasks containing 600 mL of DSM-YF (yeast free) liquid medium with glucose and incubated for up to 24 h. Biomass growth was evaluated in terms of both CFU/mL and cell concentration (cell/mL) as well as for turbidity increase (OD 600 nm). Both strains, but mainly UMB478, formed cell aggregates resulting in a 3 Log difference between colony and cell counts (Figure 1). Recently, Fuchino and Bruheim [20] highlighted that some *Z. mobilis* strains present a filamentous shape, in particular, a bulged pole as a consequence of stress or as an adaptation to the environment. Cultures reached similar OD (approximately 1.00) and cell concentrations (8.4 Log CFU/mL); as regards biomass production, both strains reached their maximum level (0.7 g CDW/L) after 8 h of incubation. These values agree with those already published (around 0.6 g CDW/L, OD close to 1.00) either for strains belonging to official collections or from mutants able to constitutively synthetize extracellular sucrose-hydrolyzing enzymes [8,21,22]. These mutants could be of interest also in the present research because of the possibility to set up biotechnological production of *Z. mobilis* biomass, employing low-cost raw ingredients (i.e., molasses).

### 3.2. Strains Sensitivity to NaCl: MIC and MBC

A disadvantage of all *Z. mobilis* strains, even for the industrial production of ethanol, is the low tolerance to inorganic ions, frequently present at inhibitory concentrations; they can be found in dough formulations (flour, water…) but also in industrial feedstocks, such as molasses and lignocellulosic residues [7], reducing the fermentation performance. One of the inorganic ions most frequently present in doughs is NaCl. To understand the sensitivity of the two *Z. mobilis* wild strains to NaCl, cell growth was studied at 10 different salt concentrations ranging from 0 to 40 g/L in cultures prepared with glucose as a carbon and energy source; in these trials, the percentage of inoculum was always maintained at 10% (*v*/*v*). The culture turbidity (OD) was measured, and the lag phase (min) and maximum growth rate (OD/h) were determined (Table 2 and Table 3). Minimum Inhibitory Concentration (MIC) and Minimum Bactericidal Concentration (MBC) were evaluated for each strain.

As for *Z. mobilis* UMB478, NaCl concentrations above 10 g/L significantly prolonged the lag phase duration (up to +75.2 ± 13.9%), while the growth rate was significantly reduced even at 2.5 g/L NaCl. The final population was reduced up to −61.0%. Both MIC and MBC were set at 20 g/L NaCl.

*Z. mobilis* UMB479 deeply modified the growth rate at 15 g/L NaCl (−46.8%). The lag phase duration significantly changed from 7.5 g/L NaCl, but with lower reduction respect to UMB478. This resistance to NaCl was also proven by the determination of MIC and MBC that were found at NaCl 20 g/L and 30 g/L, respectively.

The high physiological resistance proven by the *Z. mobilis* UMB 479 wild strain may be of interest not only in the bakery sector, in which NaCl might be present in dough formulation, but also for the commercial production of bioethanol; as *Z. mobilis* is known to be sensitive to saline stress at a mild concentration, hampering its industrial use as efficient biocatalysts, researchers have investigated adaptive evolutionary approaches to obtain salt-tolerant *Z. mobilis* strain able to growth at 10–20 g/L NaCl [20,22].

### 3.3. Doughs Production

Doughs were prepared by suspending the biomass in flour and H_2_O at a cell concentration of 5 × 10^8^ cells/g with the addition of NaCl 1.5% (*w*/*w* on flour). In this set of experiments, *S. cerevisiae* was used at the same cell concentration as *Z. mobilis* for comparison purposes. During leavening, the dough volume increased, and the total amount of produced CO_2_, the dough pH, the fermentable sugars content and the ethanol production were monitored (Figure 2).

*Z. mobilis* UMB478 produced a maximum dough volume increase of 588 ± 18 mL in 6 h of leavening, doubling the initial volume in 3.5 h. The total CO_2_ produced was only 34.6 ± 5.4 mL, and the dough pH decreased from 5.89 ± 0.00 to 5.27 ± 0.04. The data obtained by HPLC analysis confirmed the difficulty of this strain in leavening the dough (Figure 3, first row). As previously reported, this strain frequently produces cell aggregates, so far the viable count (UFC/g) was lower than the total count (cells/g): *Z. mobilis* UMB478 inoculated at 5.66 ± 0.19 Log (CFU/g) highlighted a 1 Log increase during leavening.

*Z. mobilis* UMB479 produced a high amount of CO_2_ during the leavening trials (approximately 101.0 ± 1.8 mL), doubling the initial dough volume in 2 h and reaching a final volume of 650 mL after 5 h. Glucose was almost consumed with a high amount of ethanol produced: approximately 1 g/100 g, nearly +200% respect to the ethanol produced by UMB478 (Figure 3, second row).

Using *S. cerevisiae*, the dough doubled its volume in 2 h, reaching a final level of 575 mL, lower than *Z. mobilis* UMB479 (Figure 3, third row). However, the amount of CO_2_ was the highest, reaching 132 ± 8.2 mL. As expected, baker’s yeast fermented the maltose present in the dough, consuming about 1 g/100 g during 6 h—that is half of the residual maltose present in *Z. mobilis* leavened doughs. Note that *Z. mobilis* UMB479 produced a higher and faster volume increase than *S. cerevisiae* at a similar initial cell concentration. The obtained data were higher than those reported in Musatti et al. [9] using *Z. mobilis* strains belonging to the DSMZ official collection without adding NaCl; in those trials, the lag leavening time (LLT) was in the range of 6–7 h and the leavening rate (LR) was 5–6 mL/h. Here, the two *Z. mobilis* wild strains provided faster LLT (0.6–1.2 h) and very high LR, 92–139 mL/h, confirming their interesting fermentation performance (Table 4).

Microbiological determinations relating to the beginning and end of the leavening trials are reported in Table 5. TBC values at 6 h were found significantly different between *S. cerevisiae* and *Z. mobilis* UMB478 but not with UMB479. The TBC reduction of near 0.5 Log CFU/g was also evidenced in Nissen et al. [11] employing *Z. mobilis* collection strains. Yeast and mold counts were obviously higher when the baker’s yeast was used; however, both *Z. mobilis* strains reduced the initial eumycetes naturally present in flour during dough leavening, again around 0.5 Log CFU/g. As previously reported, UMB478 accounted for a lower colony number due to its aggregation ability. The pH of the dough was found to decrease to a greater extent during *Z. mobilis* leavening (about 0.6 pH unit) compared to *S. cerevisiae*. This behavior is to be attributed to the production of organic acids by the bacterium and may be the reason for the lower TBC evidenced in these doughs.

### 3.4. Population Dynamics

The ARISA analysis is known to produce reliable correlations between microbial-community structure and environmental conditions [23,24]. ARISA was further selected for this study, as it enables a quick and cheap high-resolution fingerprint analysis, especially useful for monitoring changes in microbial communities over time for a high number of samples. Although this technique has been used in a range of studies to characterize and produce a snapshot of fungal and bacterial communities and their development within food systems [25,26], very few studies are present for dough production and focus mainly on sourdough [27]. Here, the ARISA analysis was used to understand the development of fungal and bacterial communities in doughs inoculated with *S. cerevisiae* and *Z. mobilis,* comparatively, before (t0) and after (t6) dough leavening. For each strain, two technological replicates (A and B) were analyzed.

Ecological factors and the nature of the cereals have a major impact on the establishment of a characteristic microbial community [28]. Generally, Saccharomycetales are found driving fermentation alone in culture-based methods. However, in bakery doughs, Saccharomyces has been found to be isolated at a higher frequency and together with a number of other yeasts (e.g., *Torulaspora delbrueckii*, *Kazachstania servazzii*, *Pichia fermentans*, *Wickerhamomyces anomalus*, *Kazachstania bulderi and Meyerozyma guilliermondii*) possibly in symbiotic or mutualistic interactions, probably due also to environmental and flour conditions of species diversity [29].

*Z. mobilis* is a relatively unknown bacterium in the context of dough production, with studies focusing more on the leavening performance and dough-quality assessment rather than on the bacterial community [9,11,16]. Musatti et al. [16] investigated the association of *Z. mobilis* with Fructilactobacillus sanfranciscensis; however, most community studies of association with both fungi and other bacteria involving *Z. mobilis* are still limited to the context of beverages, mainly alcoholic, in relation to fermentation and spoilage [30,31,32].

In general, fungal ARISA for doughs prepared with *S. cerevisiae* produced a higher number of species and a higher alpha diversity and evenness than those with *Z. mobilis* (*p* < 0.001) (Table 6). These high values can be attributed to the contribution of compressed yeast: it should be noted that the fungal richness and evenness of the commercial yeast preparation are considerably high (162 and 0.9, respectively). On the other hand, the low fungal richness of the doughs prepared with *Z. mobilis* may be due only to the fungal species present in the flour. After 6 h of leavening, the fungal richness remained almost constant in doughs prepared with *S. cerevisiae* and *Z. mobilis* UMB 478, while statistically decreased for *Z. mobilis* UMB 479 (*p* < 0.01).

As expected, doughs prepared with the bacterium *Z. mobilis* evidenced higher bacterial richness (max. 344) than those inoculated with compressed yeast (max. 245) (*p* < 0.001). During leavening, values remained almost constant with *Z. mobilis*, while they decreased by about 15–20% in samples produced with compressed yeast (*p* < 0.01).

Population dynamics, in terms of the number of peaks and their relative abundances (the higher the peak the higher the abundance), showed a higher number of low abundant peaks in the fungal community of doughs leavened by *S. cerevisiae* (Figure A1). Doughs prepared with *Z. mobilis,* on the other hand, showed a higher presence of highly abundant peaks. When looking at the bacterial ARISA, *Z. mobilis* showed a slightly higher portion of lower abundant peaks than *S. cerevisiae*.

*S. cerevisiae* created a more complex fungal community which showed a higher complexity from inoculum to the dough matrices, similar before and after leavening. The same was valid for the fungal community evolved by *Z. mobilis*; however, here a higher diversity was seen between the original strains and the doughs.

In terms of beta diversity, as shown by the NMDS, two main clusters occurred, for both fungal and bacterial populations of doughs, associated with the leavening agent: *Z. mobilis* or *S. cerevisiae* (Figure 4). More in-depth, Permanova analyses indicated that for *Z. mobilis*, the type of strain (*p* = 0.001) and the leavening time (*p* = 0.014) but not their interaction affected the fungal population. As regards *S. cerevisiae*, again data evidenced that the fungal community was not affected by the leavening time. The bacterial population of *S. cerevisiae* seemed to be influenced by both leavening time (*p* = 0.001), dough replicates (*p* = 0.001) and their interaction (*p* = 0.004). The bacterial population of *Z. mobilis* showed influences of leavening time, type of strain and their interaction (*p* = 0.001); pairwise comparison further confirmed differences among strains (*p* < 0.005) and between leavening times (*p* < 0.05).

Given the varied communities retrieved and the influence of the inoculum in shaping the communities, it is likely that the obtained variation across samples is the result of the interaction between a wide number of both bacterial and fungal species evolving across time together with *Z. mobilis* and *S. cerevisiae*.

## 4. Conclusions

In the present study, two *Z. mobilis* wild strains isolated from water kefir (UMB478 and UMB479) were cultured in a yeast-free liquid medium and the obtained cells were used as inoculum to produce bread-like doughs. The leavening performances and the microbial population dynamics were characterized, comparing the data with those obtained employing a commercially available baker’s yeast in compressed form (*S. cerevisiae*).

The growing performance of the two strains was similar to strains belonging to the official collection; however, UMB479 proved a very interesting resistance to NaCl (MBC 30 g/L) that may be of interest not only in the bakery sector in which NaCl might be present in dough formulation but also for the commercial production of bioethanol. Dough leavening performances were, therefore, higher than those already reported in the literature, confirming their interest in the bakery sector. The ARISA analysis, applied to understand the development of fungal and bacterial communities of doughs inoculated with *S. cerevisiae* and *Z. mobilis* comparatively, highlighted that the doughs leavened by UMB479 showed an interesting decrease in fungal richness after 6 h leavening; this feature deserves investigation as it may represent a positive aspect to be correlated with the possible increase in the shelf life of baked products.

These data will pave the way for the use of *Z. mobilis*, with particular attention to the wild strain UMB479, in bread-like formulations for the production of yeast-free baked goods.

## Figures and Tables

**Figure 1 foods-11-02768-f001:**
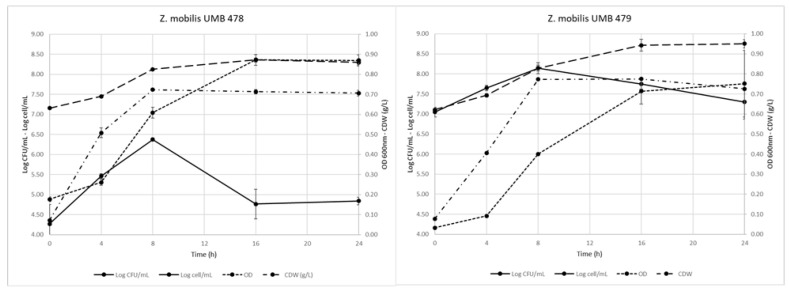
*Z. mobilis* UMB478 (**left**) and UMB479 (**right**) biomass growth evaluated in terms of colony count (Log CFU/mL), total cell count (Log cell/mL), culture turbidity (OD 600 nm) and cell dry weight (CDW g/L).

**Figure 2 foods-11-02768-f002:**
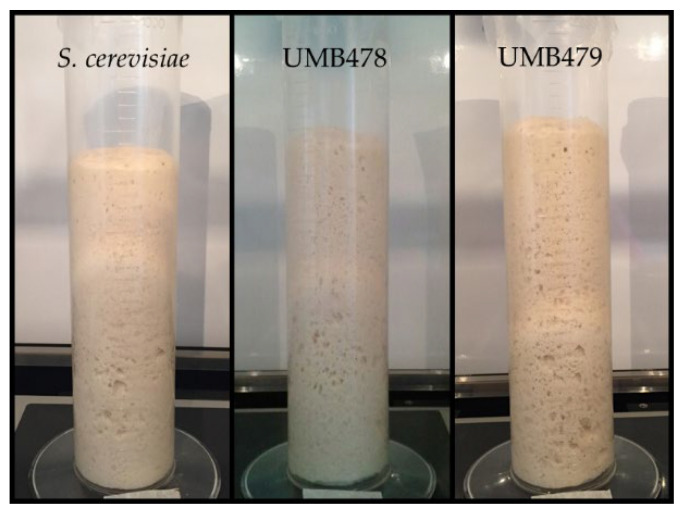
Doughs leavened (6 h) by *S. cerevisiae*, *Z. mobilis* UMB478 and UMB479.

**Figure 3 foods-11-02768-f003:**
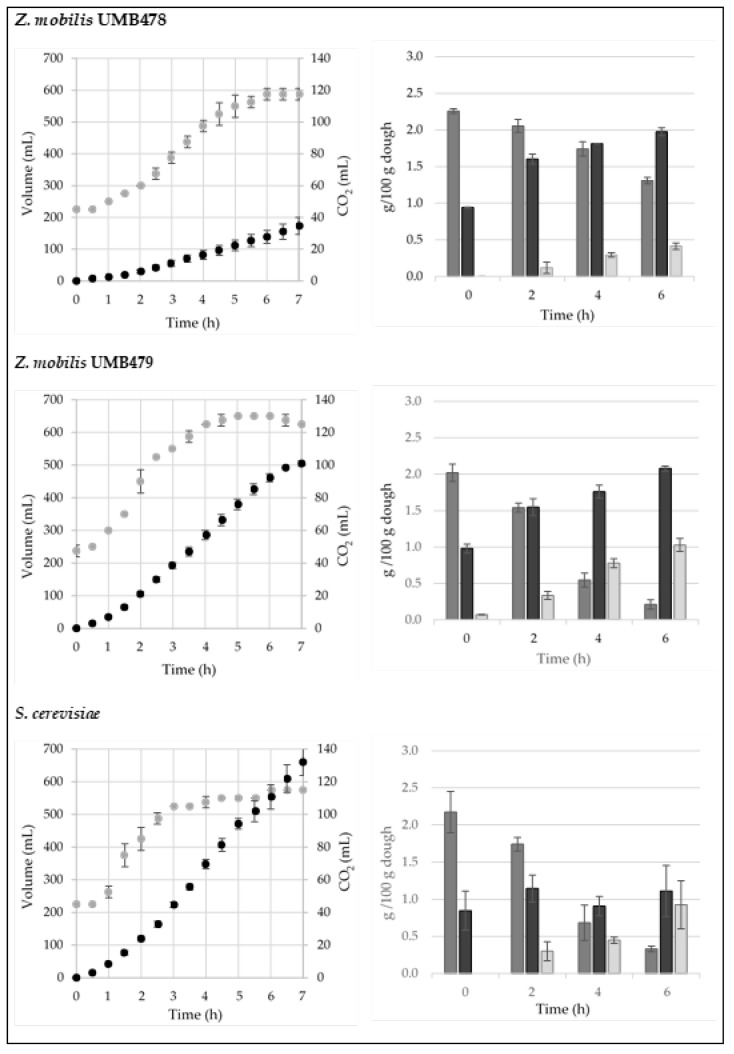
CO_2_ production (mL, black) and dough volume increase (mL, grey) of dough samples inoculated with *Z. mobilis* UMB478, UMB479 and compressed baker’s yeast (*S. cerevisiae*) (**left**); time course of glucose (dark grey), maltose (black) and ethanol (light grey) (g/100 g of dough) in dough samples (**right**).

**Figure 4 foods-11-02768-f004:**
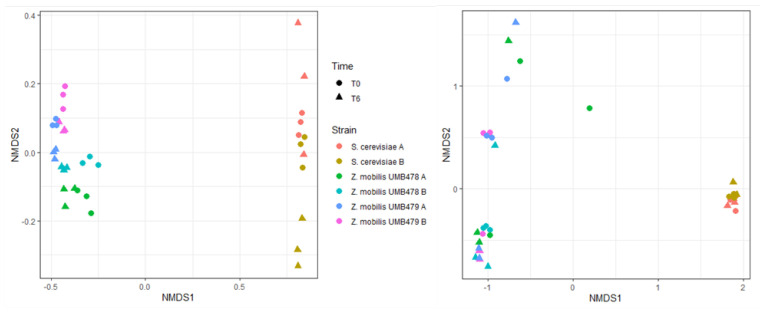
Non-metric multidimensional scaling (NMDS) of fungal (**left**) and bacterial (**right**) communities based on Bray-Curtis dissimilarities.

**Table 1 foods-11-02768-t001:** Growth curves fitting parameters of liquid cultures of *Z. mobilis* UMB478 and UMB479 containing glucose, fructose and sucrose, respectively, inoculated at 5 and 10% (*v*/*v*): lag phase duration (h), growth rate (OD/h) and final OD reached (Y_end_). Mean and standard deviation of four replicates.

Carbon Source	Strain	Inoculum (% *v*/*v*)	Lag Phase (h)	Growth Rate (OD/h)	Y_end_ (OD)
Glucose	UMB478	5	4.32 ± 0.05	0.12 ± 0.00	0.887 ± 0.008
		10	2.76 ± 0.18	0.12 ± 0.01	0.949 ± 0.035
	UMB479	5	4.47 ± 0.27	0.10 ± 0.01	0.810 ± 0.046
		10	2.60 ± 0.42	0.10 ± 0.02	0.832 ± 0.074
Fructose	UMB478	5	6.06 ± 0.20	0.09 ± 0.01	0.850 ± 0.057
		10	3.49 ± 0.18	0.09 ± 0.00	0.909 ± 0.018
	UMB479	5	5.19 ± 0.55	0.08 ± 0.01	0.815 ± 0.077
		10	4.10 ± 0.44	0.08 ± 0.00	0.736 ± 0.020
Sucrose	UMB478	5	5.14 ± 0.09	0.11 ± 0.01	0.923 ± 0.049
		10	3.51 ± 0.26	0.14 ± 0.02	1.023 ± 0.041
	UMB479	5	5.20 ± 0.17	0.11 ± 0.01	0.985 ± 0.018
		10	3.20 ± 0.34	0.12 ± 0.00	1.078 ± 0.012

**Table 2 foods-11-02768-t002:** Variations (%) in lag phase duration, growth rate and final OD reached (Y_end_) of *Z. mobilis* UMB478 grown in liquid culture in presence of NaCl (1.25–15 g/L) respect to control culture (no NaCl). Mean and standard deviation of two replicates compound; different superscript letters within a column indicate significantly different samples (*p* < 0.05).

NaCl (g/L)	Lag Phase (%)	Growth Rate (%)	Y_end_ (OD) (%)	R^2^
1.25	−1.1 ± 8.1 ^a^	−14.5 ± 4.5 ^c^	−17.5 ± 0.3 ^d^	0.995
2.5	−7.4 ± 7.9 ^a^	−25.3 ± 3.0 ^b^	−23.0 ± 0.3 ^b^	0.996
3.75	−6.4 ± 11.0 ^a^	−33.7 ± 4.0 ^b^	−18.0 ± 0.5 ^d^	0.989
5	+8.2 ± 5.9 ^a^	−21.7 ± 3.1 ^b^	−20.7 ± 0.3 ^c^	0.999
7.5	+13.7 ± 9.8 ^a^	−36.1 ± 2.7 ^b^	−23.2 ± 0.3 ^b^	0.994
10	+55.6 ± 11.3 ^b^	−37.4 ± 3.9 ^b^	−14.9 ± 0.3 ^e^	0.994
15	+75.2 ± 13.9 ^b^	−79.5 ± 1.7 ^a^	−61.0 ± 0.3 ^a^	0.994

**Table 3 foods-11-02768-t003:** Variations (%) in lag phase duration, growth rate and final OD reached (Y_end_) of *Z. mobilis* UMB479 grown in liquid culture in presence of NaCl (1.25–15 g/L) respect to control culture (no NaCl). Mean and standard deviation of two replicates; different superscript letters within a column indicate significantly different samples (*p* < 0.05).

NaCl (g/L)	Lag Phase (%)	Growth Rate (%)	Y_end_ (OD) (%)	R^2^
1.25	−8.1 ± 3.8 ^a^	−6.4 ± 3.1 ^c^	−3.9 ± 0.5 ^c^	0.998
2.5	−13.5 ± 3.6 ^a^	−12.8 ± 2.9 ^b^	−9.7 ± 0.5 ^a^	0.998
3.75	−14.4 ± 3.6 ^a^	−13.8 ± 2.9 ^b^	−10.6 ± 0.5 ^a^	0.998
5	−12.3 ± 3.7 ^a^	−7.5 ± 4.1 ^bc^	−3.6 ± 0.5 ^c^	0.998
7.5	+3.5 ± 4.3 ^b^	−6.4 ± 4.1 ^bc^	−7.2 ± 0.5 ^b^	0.997
10	+12.1 ± 4.9 ^c^	−3.2 ± 4.2 ^c^	+8.8 ± 0.7 ^d^	0.997
15	+47.0 ± 5.3 ^d^	−46.8 ± 2.2 ^a^	−5.2 ± 0.5 ^c^	0.999

**Table 4 foods-11-02768-t004:** Dough leavening properties (mean and standard deviation values of two technological replicates): lag leavening time (LLT, h), leavening rate (LR, mL/h) and maximum dough volume (DV max, mL) of dough samples leavened with the two *Z. mobilis* wild strains and baker’s yeast for comparison purposes.

Microrganism	LLT (h)	LR (mL/h)	DV Max (mL)	R^2^
*Z. mobilis* UMB478	1.2 ± 0.1	92.1 ± 12.8	586.5 ± 13.8	0.999
*Z. mobilis* UMB479	0.6 ± 0.0	139.1 ± 2.3	642.7 ± 8.2	1.000
*S. cerevisiae*	0.6 ± 0.1	153.6 ± 10.6	551.1 ± 5.2	0.999

**Table 5 foods-11-02768-t005:** Total bacterial count (TBC), yeasts and molds (Y&M) and *Z. mobilis* population (Log CFU/g) in doughs at time 0 and after 6 h of leavening with *Z. mobilis* UMB478 and UMB479 and *S. cerevisiae*. Mean and standard deviation of two replicates; different superscript letters within a column indicate significantly different samples (*p* < 0.05).

Strain	Time (h)	TBC (Log CFU/g)	Y&M (Log CFU/g)	*Z. mobilis* (Log CFU/g)	pH
*Z. mobilis* UMB478	0	5.21 ± 0.07	2.72 ± 0.09	5.66 ± 0.19	5.89± 0.01 ^c^
	6	4.71 ± 0.08 ^ab^	2.32 ± 0.19	5.65 ± 0.02	5.27 ± 0.04
*Z. mobilis* UMB479	0	5.15 ± 0.11	2.23 ± 0.14	8.41 ± 0.17	5.73 ± 0.01 ^b^
	6	4.63 ± 0.16 ^b^	1.57 ± 0.24	8.39 ± 0.19	5.14 ± 0.01
*S. cerevisiae*	0	5.28 ± 0.09	7.28 ± 0.29	-	5.31 ± 0.02 ^a^
	6	5.07 ± 0.02 ^a^	7.37 ± 0.17	-	5.13 ± 0.10

**Table 6 foods-11-02768-t006:** Species richness, diversity indices (Shannon and Simpson) and Pielou’s evenness of the bacterial and fungal communities of doughs at time 0 and after 6 h of leavening with *Z. mobilis* UMB478, UMB479 and *S. cerevisiae*. Data are indicated for both technological replicates (A and B).

Dough Sample	Fungal				Bacterial			
	Species Richness	Shannon	Simpson	Pielou’s Evenness	Species Richness	Shannon	Simpson	Pielou’s Evenness
*Z. mobilis* UMB478 A t0	106 ± 49	2.7 ± 0.5	0.8 ± 0.1	0.6 ± 0.1	274 ± 22	4.4 ± 0.1	1.0 ± 0.0	0.8 ± 0.0
*Z. mobilis* UMB478 A t6	40 ± 2	2.2 ± 0.1	0.7 ± 0.0	0.6 ± 0.0	291 ± 13	4.2 ± 0.0	0.9 ± 0.0	0.7 ± 0.0
*Z. mobilis* UMB478 B t0	63 ± 3	2.3 ± 0.0	0.8 ± 0.0	0.6 ± 0.0	278 ± 5	4.6 ± 0.0	1.0 ± 0.0	0.8 ± 0.0
*Z. mobilis* UMB478 B t6	55 ± 29	2.3 ± 0.5	0.7 ± 0.1	0.6 ± 0.0	269 ± 11	4.3 ± 0.0	0.9 ± 0.0	0.8 ± 0.0
*Z. mobilis* UMB479 A t0	70 ± 4	2.4 ± 0.1	0.8 ± 0.0	0.6 ± 0.0	344 ± 11	5.1 ± 0.2	1.0 ± 0.0	0.9 ± 0.0
*Z. mobilis* UMB479 A t6	34 ± 3	1.9 ± 0.1	0.7 ± 0.0	0.5 ± 0.0	310 ± 13	4.6 ± 0.0	1.0 ± 0.0	0.8 ± 0.0
*Z. mobilis* UMB479 B t0	68 ± 15	2.3 ± 0.2	0.8 ± 0.0	0.6 ± 0.0	313 ± 12	5.1 ± 0.1	1.0 ± 0.0	0.9 ± 0.0
*Z. mobilis* UMB479 B t6	35 ± 13	1.9 ± 0.1	0.7 ± 0.0	0.5 ± 0.0	326 ± 11	4.9 ± 0.1	1.0 ± 0.0	0.9 ± 0.0
*S. cerevisiae*	162 ± 0	4.7 ± 0	1.0 ± 0	0.9 ± 0	-	-	-	-
*S. cerevisiae* A t0	214 ± 29	5.2 ± 0.2	1.0 ± 0.0	1.0 ± 0.0	236 ± 23	4.2 ± 0.1	1.0 ± 0.0	0.8 ± 0.0
*S. cerevisiae* A t6	219 ± 19	5.2 ± 0.1	1.0 ± 0.0	1.0 ± 0.0	179 ± 17	4.1 ± 0.1	1.0 ± 0.0	0.8 ± 0.0
*S. cerevisiae* B t0	243 ± 13	5.3 ± 0.0	1.0 ± 0.0	1.0 ± 0.0	245 ± 2	4.3 ± 0.0	1.0 ± 0.0	0.8 ± 0.0
*S. cerevisiae* B t6	274 ± 29	5.4 ± 0.0	1.0 ± 0.0	1.0 ± 0.0	207 ± 15	4.2 ± 0.0	1.0 ± 0.0	0.8 ± 0.0

## Data Availability

Data are not available in public datasets; please contact the authors.

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
