# Peer review of "Characterization of Two Zymomonas mobilis Wild Strains and Analysis of Populations Dynamics during Their Leavening of Bread-like Doughs"

_foods, 2022, doi:10.3390/foods11182768_

Round 1

Reviewer 1 Report

1. The author needs to give a picture of the fermented dough.

2. The research results need to be verified on bread.

3. Photos, volume and other quality information of bread should be provided.

4. The advantages of the new strain need to be compared and analyzed.

Author Response

Reviewer 1

  1. The author needs to give a picture of the fermented dough.

Authors have added Figures:  a picture of the doughs leavened by S. cerevisiae and by the 2 Z. mobilis strains (Figure 2) and the time course of leavening performance, either in term of volume increase, CO2 developed as well as of sugars and ethanol time course during leavening (Figure 3).

  1. The research results need to be verified on bread.

The present research contains data related to preliminary trials carried out to characterize dough leavening of the 2 isolated Z. mobilis strains, in comparison with the leavening performance of S. cerevisiae. The research now is proceeding with bread production and characterization: not only the technological parameters are being characterized, but also sensory tests and volatilomic analyses will be taken into account.

  1. Photos, volume and other quality information of bread should be provided.

Authors have provided a new Figure 2 with photos of the doughs after 6 h leavening with Z. mobilis UMB478, UMB479 and S. cerevisiae comparatively. Moreover, Figure 3 was also added with the complete data on dough volume increase, CO2 developed as well as sugars consumption and ethanol production during dough leavening with the two strains of Z. mobilis and S. cerevisiae comparatively. As commented in point 2, trials are now in progress to obtain bread samples from Z. mobilis.

  1. The advantages of the new strain need to be compared and analyzed.

Section 3.1 reports on the comparison between the two studied strains in relation to biomass growth. Section 3.2 reports on the comparison between the two studied strains in relation to strain sensitivity to NaCl, that is useful when dealing with salty leavened goods. Section 3.3 reports on the comparison between the 2 Z. mobilis strains and S. cerevisiae in relation to the leaving performance of a bread-like dough formulation. Section 3.4 reports on the comparison between the dynamics of the dough population during leavening by the 2 Z. mobilis strains and S. cerevisiae. Results highlighted that one of the two Z. mobilis strains, UMB 479, offers interesting NaCl resistance and leavening performance. Authors have revised the paper trying to better highlight the distinctive features of this bacterial species and its potentiality of application in the sector of leavened baked goods.

We hope to have fulfilled all the requested revisions: all modifications are highlighted inside the manuscript.

We are looking forward to hearing from you in due course.

Yours sincerely,

                                                                                                                                     Manuela Rollini

Reviewer 2 Report

Overall – interesting study. Improvement – statistical significance throughout the study, arrangement of Table/Figures, Result & Discussion, grammar/structure

Title – Ok

Abstract – English needs a major fix. The result section needs to compare with the control (S. cerevisiae). Time needs to be uniform – 2 or 6 hours?)

Line 17 – what is the reference strain?

Introduction

Line 50 – 54 – elaborate more on the ED pathway and how this pathway can be different in zymomonas vs yeast

Line 64 – 67 – should be expanded as the paragraph looks out of place. The authors also tend to use long sentences like this throughout the manuscript. Try to shorten or divide the sentence into 2.

Include a short description of water kefir and the importance of bread population dynamics

Methods

Careful of simple grammatical mistakes e.g. line 92 (the symbol of °C)

Line 105 – why do you highlight the use of meat extract instead of yeast extract?

107 – biomass determination in a summary

Section 2.4 – does this experiment include yeast as a control? If not, how would you say that zymomonas is better than yeast?

Line 152 – why cycloheximide is used if you want to grow the yeast?

Line 152 - As you perform the population dynamics later, it seems unfair to use this compound to inhibit the growth of unwanted microorganisms. The changes in population dynamics may be caused by this compound, not the growth of zymomonas. Additionally, can the bread industry use this for consumption?

Section 2.8 – repetition of lines 133-142. Combine them.

Line 180 – why only 0 and 6 hours as the sampling is done every half an hour? Would be interesting to see what happens in between.

Section 2.9 – why ARISA is used compared to the common microbial sequencing? Any advantages?

Result and Discussion

Line 231 – use a proper subscript for Yend

Line 230 – 239 – why different growth phases were observed under different carbon sources? Elaborate

Section 3.0 – Table and Figure 1 should be here

Line 243 – would be nice if there are some pictures

Line 251 – what is the significance (in this study) of strain that can secrete sucrose-hydrolyzing enzymes?

Section 3.2 In my opinion, it is better to see the effect of inorganic ions on the leavening properties – CO2/EtOH production, dough volumes, etc under the presence of zymomonas

Table 1: significance is needed

Tables 2 and 3 – need to explain more in-text. For example, why did salt concentration at 3.75 g/L cause -33.7% lag phase but 1.25 g/L only cause -1.1%? Why there is no growth pattern in UMB479? Also, need significance

Line 301 – I think it is better to prepare a dough with varying NaCl concentrations and measure the changes

Line 327 – NDMS, TBC, LR, and LLT are not defined

Line 339 – this is an assumption?

Line 400-405 – unclear with this paragraph

Section 3.4 – you’ve established that the population dynamics are different between treatments. So, which one is suitable for bread production?

Author Response

Reviewer 2

Title – Ok

Abstract – English needs a major fix. The result section needs to compare with the control (S. cerevisiae). Time needs to be uniform – 2 or 6 hours?)

Thanks; we have revised the Abstract section trying to better focus on the obtained results and on the comparison with baker’s yeast. As regards the leavening time, S. cerevisiae was able to double the dough volume in only 2 h respect to the 6 h needed by Zymomonas; however, the final volume (the highest) reached after 6 h was reduced respect to Z. mobilis.

Line 17 – what is the reference strain?

The reference strains are those present in the official collection DSMZ, that we tested in our previous works (see Musatti et al 2020). However, inside the abstract this sentence was not important, we have decided to delete it, the suggestion of the reviewer is right.

Introduction

Line 50 – 54 – elaborate more on the ED pathway and how this pathway can be different in zymomonas vs yeast

Authors have added more info on ED pathway and on the difference with the glycolysis (lines 51-55).

Line 64 – 67 – should be expanded as the paragraph looks out of place. The authors also tend to use long sentences like this throughout the manuscript. Try to shorten or divide the sentence into 2.

This concept has been expanded and rephrased (lines 66-71). In addition, long sentences throughout the manuscript have been divided or shortened. Modifications are all easily identified with the Track Changes” function.

Include a short description of water kefir and the importance of bread population dynamics

The description of water kefir has been added (lines 73-76) as well as the importance of bread population dynamics (lines 90-91).

Methods

Careful of simple grammatical mistakes e.g. line 92 (the symbol of °C)

Thanks, we have revised and checked all the manuscript.

Line 105 – why do you highlight the use of meat extract instead of yeast extract?

The initial idea was to highlight that none of the used ingredients was derived from yeast, to consider the final dough really a yeast-free product. In any case, we have deleted this comment (line 107) to ease the sentence.

107 – biomass determination in a summary

OK, inserted (lines 112-114)

Section 2.4 – does this experiment include yeast as a control? If not, how would you say that zymomonas is better than yeast?

Sections 2.4 and 3.2 report respectively the materials and methods and the results related to trials of biomass production in liquid cultures of the 2 Z. mobilis strains, object of the research. As regards biomass production levels, no comparison was reported with S. cerevisiae because we used biomass of S. cerevisiae already grown and available on the market in form of compressed yeast. The comparison between S. cerevisiae and Z. mobilis was on their leavening performance of bread-lik doughs; these trials are described in section 2.5 (materials and methods) and section 3.3 of the results; in this case Z. mobilis UMB 479 evidenced interesting leavening performances respect to S. cerevisiae.

Line 152 – why cycloheximide is used if you want to grow the yeast?

The enumeration of the Total Bacterial Count was carried out employing the TSA culture medium; however, the yeast S. cerevisiae is able to grow on TSA, producing an overestimation of the actual bacteria; when the dough was prepared with S. cerevisiae, the medium TSA was added with cycloheximide to have the correct count of only the bacterial colonies. We have modified the description in section 2.6 related to the procedure of microbiological analyses to avoid any misunderstanding (lines 461-164).

Line 152 - As you perform the population dynamics later, it seems unfair to use this compound to inhibit the growth of unwanted microorganisms. The changes in population dynamics may be caused by this compound, not the growth of zymomonas. Additionally, can the bread industry use this for consumption?

Cycloheximmide as reported was added only in TSA solid culture medium and not to doughs, to avoid overestimation of the Total Bacterial Count due to yeast growth in the same plates. We have modified the description in section 2.6 related to the procedure of microbiological analyses to avoid any misunderstanding (lines 461-164).

Section 2.8 – repetition of lines 133-142. Combine them.

The reviewer is right: we have eliminated this paragraph and the info were combined with those already present in paragraph 2.5 (lines 136-157). The subsequent paragraphs were renumbered.

Line 180 – why only 0 and 6 hours as the sampling is done every half an hour? Would be interesting to see what happens in between.

The reviewer is right. We have added Figure 2, containing the time course (from t0 to 6 h leavening) of the volume increase, the CO2 levels and the concentration of sugars and ethanol for doughs leavened by the 2 tested strains of Z. mobilis and by S. cerevisiae.

Section 2.9 – why ARISA is used compared to the common microbial sequencing? Any advantages?

The first paragraph of section 2.8 (new renumbering) has been revised (lines 183-185), as well as section 3.4 updated to better explain why the ARISA analysis was selected (lines 407-411).

Result and Discussion

Line 231 – use a proper subscript for Yend

Thanks, modified throughout the manuscript.

Line 230 – 239 – why different growth phases were observed under different carbon sources? Elaborate.

Thanks for the suggestion, authors have commented the results (lines 241-249) taking into account the difference between glucose, fructose and sucrose uptakes in Z. mobilis. A new reference has also been inserted, thus the subsequent references’ numbers have been modified accordingly.

Section 3.0 – Table and Figure 1 should be here

OK, moved (line 266-279).

Line 243 – would be nice if there are some pictures

The reviewer is right, we have added the figure 1 (line 271).

Line 251 – what is the significance (in this study) of strain that can secrete sucrose-hydrolyzing enzymes?

Authors have added a new sentence (lines 261-264) to comment on this significance.

Section 3.2 In my opinion, it is better to see the effect of inorganic ions on the leavening properties – CO2/EtOH production, dough volumes, etc under the presence of Zymomonas

We understand the point of the reviewer. In the present manuscript, authors wanted to investigate how NaCl concentration affected Z. mobilis growth and biomass production. The new Figure 2 shows the time course of the parameters cited by the reviewer. We think that the most resistant strain able to grow at high NaCl concentration could also perform better in salty doughs. In the dough formulation used, NaCl is present at a concentration of 1.5% w/w on flour basis, that is the maximum salt level suggested for bread production in EU.

Table 1: significance is needed

Table 1 reports lag phase duration, growth rate and Yend of 2 strains grown employing 3 substrates and at 2 percentages of inoculum. We have tried to highlight all significant differences in table, but the result was difficult to understand (too many superscripts and symbols). We have added comments on significance and not-significant differences in text (lines 233-238). However, if the Editor feels for evidencing all the statistically significances, we can modify accordingly.

Tables 2 and 3 – need to explain more in-text. For example, why did salt concentration at 3.75 g/L cause -33.7% lag phase but 1.25 g/L only cause -1.1%? Why there is no growth pattern in UMB479? Also, need significance

We have added all significances in Tables 2 and 3. We have also inserted comments in text (lines 293-300). As regards the value of the lag phase at 3.75 g/L NaCl for UMB478, the reviewer is definitely right: we have copied a wrong number on the row. Please apologize.

Line 301 – I think it is better to prepare a dough with varying NaCl concentrations and measure the changes

Authors think that the most resistant strain able to grow at high NaCl concentration could also perform better in salty doughs. Nevertheless, note that in the reported trials, salt was added to flour at 1.5% w/w on flour. As bread is considered one of the most important sources of dietary salt, the EU framework for national salt initiatives to reduce salt intake has suggested the here applied concentration as the maximum to be used (Defining dietary salt and sodium - examples of implemented policies addressing salt reduction, EU).

Line 327 – NDMS, TBC, LR, and LLT are not defined

Authors have added the definitions of LR and LLT (lines 151-153); the definition of TBC has been defined in line 163; the definition of NMDS has been inserted in line 81 and explained in section 3.4 (lines 408-411).

Line 339 – this is an assumption?

Authors have modified the sentence (lines 360-364).

Line 400-405 – unclear with this paragraph

The paragraph has been rephrased for clarity (lines 442-445). Moreover, Figure 4 has been added to improve the comprehension and for clarity.

Section 3.4 – you’ve established that the population dynamics are different between treatments. So, which one is suitable for bread production?

Authors believe that the analysis of population dynamics may give an overview of what is happening into a dough from a microbial point of view when this is inoculated by S. cerevisiae or Z. mobilis. However, the technique cannot be used to decide which type of inoculum fits best; nevertheless, the interesting decrease of fungal richness evidenced after Zymomonas leavening deserves investigation and may represent a positive aspect to be correlated with the possible increase in shelf life of baked products. This comment has been inserted inside the conclusions (lines 503-505).

We hope to have fulfilled all the requested revisions: all modifications are highlighted inside the manuscript.

We are looking forward to hearing from you in due course.

Yours sincerely,

                                                                                                                                     Manuela Rollini

Round 2

Reviewer 1 Report

The author has revised it according to the review comments, which I think is acceptable.

Reviewer 2 Report

Thank you, the changes are sufficient.